# Corporate Governance Structure, Financial Capability, and the R&D Intensity in Chinese Sports Sector: Evidence from Listed Sports Companies

**Gang Chen [1],\*, James J. Zhang [2]**  **and N. David Pifer [3]**

[1] School of Economics and Management, Wuhan Sport University, Wuhan 430070, China
[2] Department of Kinesiology (Sport Management), University of Georgia, Athens, GA 30602, USA; jamesz48@uga.edu
[3] Department of Kinesiology & Sport Management, Texas Tech University, Lubbock, TX 79409, USA; David.Pifer@ttu.edu
\* Correspondence: cg224@163.com; Tel.: +86-027-87190123

**Abstract:** Innovations are the foundation of an enterprise's sustainable development, which is particularly important for sports firms in an evolving Chinese sport industrial environment. Analyzing publicly-listed sports firms on The New Third Board (NTB) in China, this study examined the influence of corporate financial capability and corporate governance structure on firms' R&D intensity through a series of multiple regression models. Findings revealed that corporate financial capability is an important determinant of R&D intensity, and corporate governance structure has a small but meaningful effect on R&D intensity. Specifically, for Chinese sports firms, several financial capability indicators, such as return on equity, accounts receivable turnover, assets turnover, and profit growth rate, have positive relationships with R&D intensity; however, other financial capability indicators, such as leverage and cash flow, have negative relationships with R&D intensity. Limited evidence was found to support the notion that corporate governance significantly influences R&D intensity, although sports firms with good governance mechanisms are more likely to increase the positive effects of financial capabilities on R&D intensity while decreasing the negative effects. Discussions were centered on planning and executing R&D activities in sports companies.

**Keywords:** corporate financial capability; corporate governance structure; listed sports enterprises in new three-board in China; R&D intensity; policy effectiveness

## 1. Introduction

Social, psychological, economic, and environmental impacts of organized sports activities and sports events are well-documented in previous studies [1]. When planned and managed effectively, sports can be a green industry that positively promotes the public welfare of communities and citizens. Hence, the sports industry is a green industry that is very important for the sustainable development of mankind. However, there is still a gap between the sports industry in China and that of some western countries. Due to growing interest in leisure activities, adoption of healthy lifestyles, augmentation of sports competitions and events, and the evolution of sports media technology, the sports industry has experienced rapid growth and has become one of the largest industries in some western countries. For example, the estimated size of the sports business industry in USA has risen sharply to approximately $498.4 billion in 2015 [2]. Sports leagues, teams, and events, the most popular leisure and entertainment options, have been integral to many communities' cultural and economic foundations in western countries [3]. However, such strong market demand and such competitive

sports firms in China lacked before 2014, and the total output of the sports industry in China was only 1.1 trillion RMB in 2013. Meanwhile, the international nature of modern sport requires sport organizations to modify their management practices in order to remain effective and competitive in a border-transcended marketplace [4]. Differences in such areas as culture, policy and regulation, language, and the environment in global, national, regional, and local communities make this a challenging task. This requires that continual innovation is implemented to promote the rapid and sustainable development of the Chinese sports industry.

In order to rapidly develop the sports industry, the central government of China released Document No.46 titled "to rapidly develop sports industry and promote sports consumption" in 2014. Under its support, China's sports sector has been growing at an annual rate of over 10% in recent years, with some sectors, such as sports facility construction, sport management services, and fitness and leisure, growing at a rate of over 50%. The sports industry in China has become a veritable rising industry. Obviously, the role of innovation of sports firms in the development of China's sports industry cannot be overlooked. Especially, innovation-driven policies were also introduced in Document No. 46. Partially attributing to this market environment, numerous new start-ups have emerged in the sports industry, and more and more of them have begun to put emphasis on nurturing the capacity of enterprise innovation. A large amount of literature also showed that many national or local governments routinely implement innovation-driven policies or other forms of stimulation to address hindrance associated with R&D underinvestment and stimulate creativity, advancement, and competitiveness, such as tax relief and deductions, R&D subsidies, and regulatory provisions [5,6].

Numerous researchers have indicated that research-based innovations are the foundation for a business corporation to survive, thrive, and sustain; sports businesses are no exception. Research and Development (R&D) has become a core business component of firms [7] as it is both a key ingredient in the introduction of new products and processes [8] and a key source of rapid growth, sustainable development, and competitive advantages [9]. R&D investment plays a decisive role in initiating and sustaining innovation activities of technological enterprises [10], not only improving short-term and long-term financial performance but also facilitating the acquisition of competitive advantages [11]. Thus, it is very necessary to focus on the innovation development of sports firms, especially their R&D investment issue. Choosing the level of R&D as the criterion variable and identifying its predicting variables in this study would be ultimately promoting the sustainable development of sports enterprises in China.

Although the Chinese government has promoted a creative business cultural environment through the innovation-driven policies in an effort to advance the development of the sports industry in China, an innovation-driven policy is merely an external incentive to affect R&D investment. In the meantime, the sustainable development of a sports enterprise would greatly depend on many of its internal factors in finance and governance. An enterprise's financial capability has been recognized by previous researchers as the most pertinent determinant of its R&D investment, which can be facilitated or even deterred by the firm's governance structure and its subsequent decision-making on R&D investment. As a rising industry, there are numerous new start-ups emerging in the sports industry in China. Yet, smaller, newer, and more technology-intensive firms face financial and corporate governance constraints [12,13] despite recognizing the importance and relevance of research and development activities for the firm's long-term well-being [14]. Today, many sports enterprises face financing and corporate governance constraints, which have hampered their market competitiveness, highlighting the significance of examining the influence of these concepts on research and development. Therefore, the purpose of this study was to examine the impact of corporate financial capacity and governance structure on the level of R&D investment of sports firms in the evolving and growing economic environment of China.

Based on the above literature, and considering that China's sports industry is an emerging industry with smaller and younger firms, this study examined the impact of corporate financial capability and governance structure on firms' R&D intensity by analyzing sports firms listed on The New Third

Board. Here, R&D intensity is defined as R&D expenditure divided by total assets [15–18]. This study employed multiple linear regression models to answer the following three research questions:

1. What is the relationship between corporate financial capability and R&D intensity in Chinese sports firms?
2. How do corporate governance structures affect R&D intensity in Chinese sports firms?
3. Are the effects of corporate financial capability on R&D intensity in Chinese sports firms different for firms with different governance structures?

This study was focused on the effects of corporate financial capability and governance structure on R&D investment in the context of the Chinese sports industry. Findings of this study potentially have the following implications pertaining to planning and executing R&D activities within sport companies: (a) identifying the importance and relevance of corporate financial capacity would help sport companies make appropriate decisions on investing into R&D activities and making choices among such options as self-innovation, purchasing intellectual property rights, and affiliating with larger corporations; (b) adopting a feasible governance structure would enhance the effectiveness of investing into and operating of R&D activities; (c) by exploring the interaction effect of financial capacity and governance structure on R&D intensity, suggestions can be made for corporations with different financial capacity to adopt differential governance structures in an effort to enhance their innovative activities and market competitiveness to get ready for the ever-changing market environment; and most importantly; (d) this study paid a particular attention to the enterprise innovation in a newly developing industry in a growing economy. Although the research sample comes from the Chinese sports industry, the research conclusion has some reference value for the innovation development of sports firms in other nations or geographical locations, even for the innovation development of new start-ups or new industries.

In the sections that follow, the theoretical framework and hypotheses have been developed to account for prior studies, exploring (a) the effect of corporate financial capability on R&D intensity, (b) the effect of corporate governance structure on R&D intensity, and (c) the interaction effect between corporate financial capability and governance structure on R&D intensity. Next, the method has been introduced, followed by an examination of the results. We concluded with a thorough discussion of the implications of the study based on related theories and practices.

## 2. Theoretical Framework and Hypothesis Development

Our theoretical framework is presented in Figure 1, and the formulation of the study hypotheses is laid out in the subsequent sections.

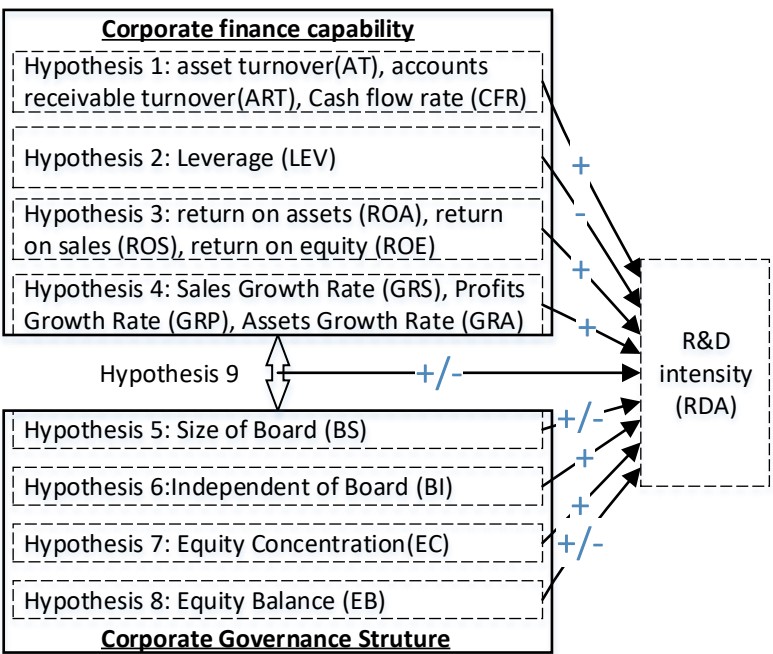

**Figure 1.** Theoretical Framework.

*2.1. Corporate Financial Capability Impacting R&D Investment*

As R&D activities are generally long-term and high-risk, returns on their investment are unpredictable, and sometimes, their results are intangible capital without collateral function [19,20] and require business secrets and core technologies to remain undisclosed [21]. They may also quickly become obsolete, devaluing their potential returns [22]. As such, firms have difficulty funding R&D activities by using external financing channels [16,20,23]; even if firms' R&D activities receive external, indirect support through financing or loans, this depends on their financial health. Thus, a firm's innovation is significantly correlated with its cash holding ratio, sales volume, and profitability [17]. R&D investment also depends on cash flow or stock issues and is constrained by external financing resources [16]. Here, liquidity, solvency, profitability, and value-added are the four basic elements of financial health [24–26]. Corporate financial capabilities, including solvency, profitability, operational capability, and growth capability [27], have been identified as factors influencing R&D investment [28,29]. In the sections that follow, financial health elements are examined as they relate to R&D intensity in prior studies.

**Liquidity, firms' operating capabilities, and R&D investment.** A firm's financial condition and flow of assets have an impact on its R&D expenditures [30]. The effect of asset liquidity on innovation is larger for smaller and younger firms [10]. Additionally, internal cash flows are the main source of R&D investment [16], and R&D intensity is positively related to cash flow [31,32]. There are three important firm-level indicators for asset liquidity and internal cash flow: (a) the total assets turnover ratio (AT), defined as the ratio of sales to average total assets; (b) accounts receivable turnover ratio (ART), defined as the ratio of sales to average accounts receivable; and (c) cash flow rate (CFR), defined as the cash flow from operations scaled by total assets [16,17,33].

**Firms' solvency capabilities and R&D investment.** Smaller and younger firms face higher financial constraints [9], and the effect of financial constraints on their R&D investment is more pronounced [23]. R&D investment is negatively related to debt [34] and debt financing [35]. Leverage (LEV) is defined as total debt scaled by total assets [33,36] and can reflect the long-term solvency of firms. LEV is negatively related to R&D investment [37] and influences a firms' ability to achieve liquidity [38].

**Firm profitability and R&D investment.** The three most popular financial performance indicators used to measure current profitability are return on assets (ROA), return on sales (ROS), and return on equity (ROE) [26,28]. ROA is defined as the ratio of net income to total assets [10,33]; ROS is defined

as the ratio of net income to sales [26,28]; ROE is defined as the ratio of net income to shareholders' equity [17,28]. The corporate R&D expenditure is relative to corporate profitability; but, it is not significantly related to abnormal returns of firms in American bio-pharmaceutical listed companies [39].

**Value-added, firm growth, and R&D investment.** Output or value-added is the main explanatory variable for R&D [40]. As an important outcome of the proactive management of R&D expenditures [41], growth is the primary goal of firms and can indicate market valuation and investment opportunities [42,43]. Firms with better investment opportunities usually pay closer attention to investments in R&D [42,43]. The growth of firms is closely related to R&D investment [44]. The stronger the growth capability of a firm, the higher its R&D investment is likely to be [41]. Similar to Kuo et al. (2018) [28], Mudambi and Swift (2011) [45], and Patel, Guedes, Soares, and Gonçalves (2018) [46], the output variable shows the value-added from growth rate of sales (GRS), growth rate of assets (GRA), and growth rate of profit (GRP).

Based on the aforementioned theoretical underpinnings, the following hypotheses are developed in an attempt to answer the first research question:

RQ1—What is the relationship between corporate financial capability and R&D intensity in Chinese sports firms?

**Hypothesis 1.** *AT, ART, and CFR are positively related to R&D intensity.*

**Hypothesis 2.** *LEV is negatively relative to R&D intensity.*

**Hypothesis 3.** *ROS, ROA, and ROE are positively related to R&D intensity.*

**Hypothesis 4.** *GRS, GRA, and GRP are positively related to R&D intensity.*

*2.2. Corporate Governance Structure Impacting R&D Investment*

Corporate governance is an institutional form that aligns the interests of owners and managers. It often provides monitoring arrangements for owners or an incentive system to persuade managers to take risks [47]. Because the activities of R&D investment are carried out under the established corporate governance structure, corporate governance plays an important role in R&D investment [47], and corporate governance structure fundamentally affects firm R&D investment [4,42,48,49]. Corporate governance structure (i.e., the top management, the board, and the structure of the shareholders) has been identified as a key determinant of R&D investment decisions [4]. Specifically, an effective board facilitates R&D investment and reduces R&D's sensitivity to cash flow [34]. The board and ownership structure also affect R&D activities [50].

Board size (BS) and board independence (BI). As the twin cores of corporate governance structure, the size of the board (BS) and the independence of a board (BI) affect the heterogeneity of the top management team [33,51], thereby affecting R&D investment decisions [52]. BS is defined as the total number of directors on board [33], and BI is defined as a percentage of the independent directors on the board [4,33,51]. Whether BS and BI can incentivize managers to promote R&D remains a subject of debate [53]. Some scholars think that an independent board with a supermajority of independent outsiders is a mainstay of effective corporate governance [53–55] and that BI is positively correlated with R&D investment [56]. However, others think that independent directors lack inside information [57], and the monitoring function of independent directors in inducing managers' greater managerial efforts and risk-aversion are often neglected [58]. When the size of a board increases, the supervision and motivation from board to manager shrinks [59]. Therefore, BS and BI play a unique and complex role in R&D investment decisions.

Degree of equity concentration (EC) and the degree of equity balances (EB). According to Jensen and Meckling (1976) [60], an important prerequisite of the sound development of firms is that their equity is moderately concentrated, and the major shareholder has neither dictatorial power over excessively concentrated equity nor passive decision for excessively scattered equity. An excessively

large shareholding ratio of the first major shareholder has a negative effect on technological innovation and corporate performance [33,51]. The shareholding ratio of the first major shareholder is used to measure the degree of equity concentration [21,36]. The equity structure without effective supervising and balances could lead to one-sided decision-making, and it may even damage the interests of minority shareholders. The degree of equity balances is measured by the shareholding ratio sum from the second to fifth shareholders.

Based on the above analyses, corporate governance structure affects R&D investment. Meanwhile, the corporate governance structure also affects corporate finance. For example, the concentration of equity has an inverted U-shaped relationship with ROE [61]. Moreover, financial constraints would disappear in well-governed firms [62]. Vo and Le (2017) [17] considered that the effects of corporate governance structure on R&D investment are driven by certain financial indicators, such as profitability and cash flow. This leads to the following hypotheses for the second research question:

RQ2—How do corporate governance structures affect R&D intensity in Chinese sports firms?

**Hypothesis 5.** *A moderate BS is positively related to R&D investment in sports enterprises.*

**Hypothesis 6.** *BI is positively related to R&D investment.*

**Hypothesis 7.** *EB positively impacts R&D investment.*

**Hypothesis 8.** *EC significantly influences R&D investment.*

*2.3. The Interaction Effect between Corporate Financial Capability and Corporate Governance Structure*

The effects of corporate governance structure and corporate financial capability happen throughout the entire R&D process and occur at all levels of firm strategy decision-making. Moreover, they are not independent of one another, and their interaction can have a complex impact on a firm's R&D intensity. Driver and Guedes (2012) [47] argued that this interactive effect of governance and cash flow on R&D has not previously been investigated in R&D investment studies, and Vo and Le (2017) [17] echoed that the effects of corporate governance structure on R&D investment are driven by some financial capability indicators, such as profitability and cash flow. Thus, the effect mechanism of governance structure on R&D intensity constitutes a complex problem. Conditions vary across industries [63], and the demands and competition of each industry may affect the level of R&D investment [63,64]. As such, the heterogeneity of firms must be accounted for through the interaction effects of corporate governance structure and corporate financial capability [28].

Different from the firms that have strong, established governance systems, weakly governed firms spend more on the acquisition, do not utilize cash internally, and disperse their cash reserves quickly, so they have fewer amounts of cash in reserve. Moreover, they often have low insider ownership, fewer shareholder rights, and independent boards so that future investment value and profitability decreases [36]. Director ownership and board size are negatively related to corporate cash holdings [36]. The concentration of shares and ownership percentage of the five big shareholders are directly related to cash holdings [36]. The size of the board has a negative impact on the R&D investment and cash flow relationship [33] and is positively related to leverage for the US firms [65].

For external or internal finance suppliers, corporate governance is an important way to assure their return on investment [66]. Thus, well-governed firms can obtain more support from finance suppliers. Moreover, under the monitoring and control of corporate governance mechanisms, the managers do not tend to misuse the cash [36], and their self-dealing activities are confined [66]. A more independent board encourages better resource allocation when free cash flows are paid back to owners [67]. Inside directors facilitate better resource allocation when their firms face rich growth opportunities [67]. Thus, the independence of a board tempers the relationship between growth prospects and R&D intensity [67].

According to previous research findings, mutual monitoring mechanism provides important checks and balances on CEO's power and can mitigate the agency problem [68], improve the value of the firm, and reduce the CEO's ability to pursue "quiet life" within a given context of corporate governance structure [69]. Moreover, higher fractional ownership, more optimal equity incentive levels, and more new equity incentives can lead to better firm performance [70]. Weak governance firms in non-competitive industries have lower equity returns, worse operating performance, and lower firm value. Weak governance firms in noncompetitive situations have lower labor productivity and higher input costs than do good governance firms [71]. To a varying degree, firm performance, market uncertainty, and investment risk of firms are associated with return-on-investment [72]; in the meantime, credit risk-adjusted inside the debt often motivates executives to be conservative [60], which can be especially heightened when a credit crisis occurs in the marketplace [73]. The internal pay differential has a positive relationship with firm risk, R&D intensity, firm focus, and financial leverage [74]. The industry pay gap affects firm performance, risk, investment policy, and financial policy, and the magnitude of its effects varies across industries and over time [75]. These lead to Hypothesis 9 after the third research question:

RQ3—Are the effects of corporate financial capability on R&D intensity in Chinese sports firms different for firms with different governance structures?

**Hypothesis 9.** *The interaction between corporate financial capability and corporate governance structure significantly affects sports firms' R&D intensity.*

## 3. Materials and Methods

### 3.1. Research Variables

R&D intensity is a measure for the effort the company puts into pursuing innovation and can also be used as proxies for the absorptive capacity of the firm and measures to rank companies and countries by government entities [76]. Considering that total assets are the most important factor influencing R&D investment and an important determinant of R&D ability, following related studies [10,16,52], we selected R&D intensity (RDA) as our dependent variable and measured it by using R&D expenditure divided by total assets. Following the convention, all missing values for R&D expenditure were replaced with zeroes, and the upper limit of R&D intensity was set at 1 [77]. There are several factors that influence R&D investment. Standing from the perspective of firms—financial capability and corporate governance are important factors influencing R&D investment. This study selected the independent variables, including the indicators of corporate financial capability, namely ROA, ROS, ROE, GRS, GRP, GRA, AT, ART, CFR, LEV, and the indicators of corporate governance structure, namely EB, EC, BS, BI. Table 1 summarizes the variables included in the current study.

**Table 1.** Indicator of corporation financial capability, governance structure, and R&D activity.

| Variables (Abbreviation) | | | Measurement |
|---|---|---|---|
| | R&D intensity (RDA) | | The ratio of R&D expenditures to total assets |
| Corporate governance structure | Equity concentration (EC) | | The shareholding ratio of the first major shareholder |
| | Equity balances (EB) | | The shareholding ratio sum from the second to fifth shareholders |
| | Size and independence | Size of the board (BS) | All the total numbers of directions on the board |
| | | Independence of board (BI) | A percentage of the independent directors on the board |
| Corporate financial capability | Profitability | Return on assets (ROA) | The ratio of net income to total assets |
| | | Return on sales (ROS) | The ratio of net income to sales |
| | | Return on equity (ROE) | The ratio of net income to equity |
| | Growth | Sales growth rate (GRS) | The annual percentage growth in sales from the previous year |
| | | Profits growth rate (GRP) | The annual percentage growth in profits from the previous year |
| | | Assets growth rate (GRA) | The annual percentage growth in assets from the previous year |
| | Operating | Asset turnover (AT) | The ratio of sales to average total assets |
| | | Accounts receivable turnover (ART) | The ratio of sales to average account receivable |
| | | Cash flow rate (CFR) | The ratio of net cash flows generated in business activities to initial total assets |
| | Solvency | Leverage (LEV) | The ratio of total debt to total assets |

## 3.2. Empirical Models

RQ1 explores the key indicators of corporate financial capabilities that determine a sports firm's R&D intensity, which in turn reveals the effect of corporate financial capability on R&D investment. To examine Hypotheses 1–4, the following regression model was developed based on the above literature reviews:

$$
\begin{aligned}
RDA = \quad & \lambda_1 ROA + \lambda_2 ROS + \lambda_3 ROE + \lambda_4 GRS + \lambda_5 GRP + \lambda_6 GRA \\
& + \lambda_7 AT + \lambda_8 ART + \lambda_9 CFR + \lambda_{10} LEV + \delta
\end{aligned}
\tag{1}
$$

In Equation (1), *RDA* is the R&D intensity of a sports firm, as measured by R&D expenditures divided by total assets; *ROA*, *ROS*, and *ROE* are the profitability indicators of firms; *GRS*, *GRP*, and *GRA* are the growth indicators of firms in the observed year; *AT*, *ART*, and *CFR* are the operating capability indicators of firms; *LEV* is the solvency indicator of firms.

RQ2 explores the effect of corporate governance structure on the R&D intensity of sports firms. An examination of RQ2 translates to the following two regression models:

$$
RDA = \gamma_1 EC + \gamma_2 EB + \gamma_3 BS + \gamma_4 BI + \gamma
\tag{2}
$$

In Equation (2), *BC* is the shareholding ratio of the biggest shareholder; *EB* is the sum of shares held by the second to fifth largest shareholders; *BS* is all the total numbers of directions on the board; *BI* is a percentage of the independent directors.

Equation (2) shows that corporate governance structure directly affects R&D intensity, demonstrating the compound effect of both corporate financial capability and corporate governance structure on the decision of R&D investment. The exploration of RQ1 and RQ2 led to the development of the following regression model.

$$\begin{aligned} RDA = \quad & \lambda_1 ROA + \lambda_2 ROS + \lambda_3 ROE + \lambda_4 GRS + \lambda_5 GRP + \lambda_6 GRA \\ & +\lambda_7 AT + \lambda_8 ART + \lambda_9 CFR + \lambda_{10} LEV + \gamma_1 EC + \gamma_2 EB + \gamma_3 SB + \gamma_4 IB + \gamma + \delta \end{aligned} \qquad (3)$$

RQ3 explores the heterogeneous effects of corporate governance structure on R&D intensity. Equation (4) is utilized to address RQ3. Equation (4) includes the interaction effects between the key indicators of corporate governance structure and the key indicators of corporate financial capability.

$$RDA = [\delta][Fin] \times [Gov] + \vartheta \qquad (4)$$

In Equation (4), $[Fin]$ are the key indicators of corporate financial capability significantly related with R&D intensity based on the examination of the primary regressions (1) and (3). $[Gov]$ are the key indicators of corporate governance structure significantly related with R&D intensity based on the examination of the primary regressions (Equations (2) and (3)).

### 3.3. Research Context

The sports industry is a newly rising business sector in China. Since the announcement of "Document No. 46" in 2014, increasing social capital has flown into sports business, and more and more sports firms have begun to be listed on The New Third Board. Visions, strategies, and operational procedures for entrepreneurship and innovation need to be developed and carried out by these new corporations to break governance barrios, overcome financial constraints, and sustain market competitiveness. In order to understand the mechanism of enterprise innovation in a new sports industry environment, this study focused on Chinese sports firms listed on The New Third Board and examined the effect of corporate governance structure and financial capability on organizational R&D intensity.

Different from the Shanghai and Shenzhen Stock Exchanges, The New Third Board is a stock exchange market established in 2006, targeting start-up firms with lower listing requirements of the corporation's assets scale. Usually, these businesses are relatively concentrated with their production, services, and even geographical locations. While a majority of sports firms listed on The New Third Board mainly focus on sports products and services, some are sub-sets of larger companies also having ventures in other business sectors. In an effort to include all sports enterprises that are of single or multiple product(s) and services, as well as those being a sub-division of a larger corporate, this study first used www.baidu.com to search for financial report, news, or comments with "Sport concept stock on The New Third Board" as topic and recognized sports firms to the greatest extent based on the content analysis of these documents; additionally, financial reports of firms were made based on their reports following the standard of "National Statistical Classification of Sports Industry" (Order No. 17 of the National Bureau of Statistics of the People' s Republic of China, 2015) [78] to select corporations with sports business segments.

The sample data came from the 2017 annual report of sports firms listed on the new enterprise board in China, which was actually publicized in 2018. There were a total of 95 related sports firms listed on The New Third Board in China in 2017. Of them, 12 firms did not release their annual reports on time and could not be included in the study. Consequently, 83 firms remained and were included in the analyses in this study. Although the size of research samples might not appear large, the study included all of the sports firms listed on The New Third Board, representing the entire population of the firms in actuality. According to Pearson et al. (2003), data analyses at the population parameter level would substantially reduce errors in statistical inferences [79]; in fact, when the entire population was involved, the sample size was no longer the main concern. Considering that there were 10 independent variables in this study, the sample size was close to meet the 10:1 ratio criterion, as suggested by Hair et al. (2010) [80]. More importantly, among these 83 firms were mainly sporting goods manufactures, sports field construction companies, e-sports companies, sports media, or sports

software development companies that heavily relied on entrepreneurship, technology, and innovation for development and sustainability. These accounted for 82% of the enterprises included in the study. In addition to these firms, there were three sports clubs, seven health-fitness centers, and five sports equipment sales companies, which had customer service as their primary business practices, which might be of different nature and process in terms of innovation activities when compared with sports companies specializing in manufactures and productions. For example, in an effort to innovate advance customer services, one health-fitness center provided sports medicine care, and a sport equipment sales firm offered technical support to customers. It is also necessary to note that among the firms included in the sample, there were 59.04% of sports firms belonging to the high-tech enterprise; this further showed that the sports firms played an important role in the innovation development of the sports industry. Table 2 summarizes key firm variables extracted from the annual reports.

**Table 2.** Descriptive statistics for the corporate financial capability and corporate governance structure variables.

| Variables | Minimum | Maximum | Mean | Std. Deviation |
|---|---|---|---|---|
| RDA | 0 | 0.44 | 0.0405 | 0.05822 |
| ROS | −1.87 | 1.02 | 0.0121 | 0.34992 |
| ROE | −8.67 | 1.11 | −0.1850 | 1.33909 |
| ROA | −0.97 | 0.62 | 0.0039 | 0.24605 |
| LEV | 0.01 | 2.59 | 0.4515 | 0.34218 |
| CFR | −1.27 | 0.63 | 0.0041 | 0.23469 |
| ART | 0 | 277.14 | 15.5607 | 41.51761 |
| AT | 0.15 | 3.14 | 1.1407 | 0.71957 |
| GRA | −0.62 | 3 | 0.4587 | 0.71596 |
| GRS | −0.59 | 18,524.68 | 223.5341 | 2033.31015 |
| GRP | −28.26 | 11.20 | −0.4737 | 4.66542 |
| EC | 0.20 | 0.99 | 0.5195 | 0.18665 |
| EB | 0.01 | 0.72 | 0.3608 | 0.14349 |
| BS | 5 | 12 | 5.6867 | 1.33369 |
| BI | 0 | 0.40 | 0.0209 | 0.08325 |

According to Table 2, the averages of ROS, ROE, and ROA were, respectively, 0.0121, −0.1850, and 0.0039. The mean leverage was 0.4515, which was an appropriate level in the range between 0.4 and 0.6. The averages of GRA and GRS were large, but the deviation of GRS was very large, and the average of GRP was less than 0. This was because most sports firms' profits were decreasing, and individuals had a very high sales growth rate. The average of EC was very large and more than 0.5, but the average of EB was relatively little. That is, most sports firms had a higher degree of equity concentration and a lower degree of equity balances. The average of BS and BI were 5.6867 and 0.0209, respectively, and their deviations were rather small. This was because most sports firms (72.3%) only had five directors, and only very few firms (6.02%) had an independent director. This meant that most sports firms' governance structures were very simple and lacked openness and variety.

## 4. Results

RQ1 explored which indicators of corporate financial capability were associated with their R&D intensity. Multiple regression analyses were conducted to examine the research hypotheses and questions. The first and third columns of Table 3 show the results of model 1 and model 3, which tested H1, H2, H3, and H4 (i.e., corporate financial capabilities would be significantly associated with firms' R&D intensity). The results of model 1 indicated that corporate financial capability directly explained 50% of the variance in firms' R&D intensity ($F_{(10, 72)} = 7.201$, $p < 0.01$). Meanwhile, the results of model 3 indicated that corporate financial capability and corporate governance structure together explained 53.1% of the variance in firms' R&D intensity ($F_{(14, 68)} = 5.489$, $p < 0.01$). However, whereas R&D intensity was positively related to ROE, it was not significantly related to ROS and ROA; thus, H1

was partially supported. R&D intensity was negatively related to LEV; thus, H2 was supported. R&D intensity was positively related to ART and AT but negatively related to CFR; thus, H3 was partially supported. Finally, H4 was partially supported as R&D intensity was positively related to GRP but not significantly related to GRS or GRA.

**Table 3.** Regression analyses of R&D intensity based on corporate financial capability and corporate governance structure.

| Variable | Abbreviation | Model 1 | Model 2 | Model 3 |
|---|---|---|---|---|
| | (Constant) | 0.039 *** (0.013) | 0.065 (0.071) | −0.008 (0.057) |
| Profitability | ROS | −0.019 (0.028) | | −0.002 (0.032) |
| | ROE | 0.013 ** (0.006) | | 0.012 ** (0.006) |
| | ROA | −0.082 (0.052) | | −0.085 (0.053) |
| Solvency | LEV | −0.057 *** (0.020) | | −0.043 ** (0.021) |
| Operating | CFR | −0.145 *** (0.032) | | −0.147 *** (0.032) |
| | ART | 0 ** (0) | | 0 ** (0) |
| | AT | 0.029 *** (0.007) | | 0.028 *** (0.007) |
| Growth | GRA | −0.011 (0.008) | | −0.010 (0.008) |
| | GRS | −0 (0) | | −0 (0) |
| | GRP | 0.005 *** (0.001) | | 0.005 *** (0.001) |
| Governance | EC | | −0.041 (0.058) | 0.036 (0.047) |
| | EB | | 0.055 (0.073) | 0.104 * (0.060) |
| | BS | | −0.004 (0.006) | −0.002 (0.005) |
| | BI | | −0.056 (0.086) | 0.004 (0.071) |
| F-value | | 7.201 *** | 1.642 | 5.489 *** |
| R-squared | | 0.500 | 0.078 | 0.531 |

Notes: Dependent variable is RDA; the standard errors of the parameter assessments are included in brackets (); * $p < 0.1$, ** $p < 0.05$, *** $p < 0.01$.

RQ2 explored how corporate governance structure affected R&D intensity in Chinese sports firms. The second and third columns of Table 4 tested if corporate governance structure was significantly associated with firms' R&D intensity. Model 2 explored the effect of corporate governance structure on R&D intensity, and its *p*-value was not only greater than 0.05 but also greater than 0.1 (i.e., $p > 0.1$). It indicated that the corporate governance structure was not directly affecting R&D intensity. However, referring to the research of Alam, Uddin, & Yazdifar (2019) [81] and Zhang & Guan (2018) [82] et al., although a *p*-value was only less than 0.1, some new details should be neglected as following: the result of model 3 ($F_{(14, 68)} = 5.489$, $p < 0.01$) showed that R&D intensity did not have a significant relationship with EC, BS, and BI ($p > 0.1$), but it had a weak, positive relationship with EB ($p < 0.1$). Thus, H7 was weakly supported, but there was no evidence to confirm H5, H6, and H8. Variance explained in model 3 was more than that in model 1, further suggesting that corporate governance

structure had a small but meaningful impact on R&D intensity, which occurred in the presence of corporate financial capability.

**Table 4.** Regression analyses based on R&D intensity and the interaction between corporate financial capability and corporate governance structure.

| Variable | | Model 4 | | Model 5 | | Model 6 |
|---|---|---|---|---|---|---|
| | (Constant) | 0.019 * (0.010) | (Constant) | 0.050 ** (0.012) | (constant) | 0.048 *** (0.012) |
| Profitability | ROS*EB | 0.003 (0.064) | ROS*EC | −0.068 (0.012) | ROS*BS | 0 (0.004) |
| | ROE*EB | 0.029 ** (0.014) | ROE*EC | 0.032 ** (0.013) | ROE*BS | 0.002 ** (0.001) |
| | ROA*EB | −0.125 (0.115) | ROA*EC | −0.217 * (0.121) | ROA*BS | −0.018 * (0.009) |
| Solvency | LEV*EB | −0.019 (0.057) | LEV*EC | −0.129 *** (0.047) | LEV*BS | −0.011 *** (0.003) |
| Operating | CFR*EB | −0.370 *** (0.072) | CFR*EC | −0.223 *** (0.067) | CFR*BS | −0.026 *** (0.006) |
| | ART*EB | 0.001 ** (0) | ART*EC | 0.001 ** (0) | ART*BS | 0 ** (0) |
| | AT*EB | 0.067 *** (0.017) | AT*EC | 0.052 *** (0.015) | AT*BS | 0.004 *** (0.001) |
| Growth | GRA*EB | −0.024 (0.017) | GRA*EC | −0.019 (0.016) | GRA*BS | −0.003 ** (0.01) |
| | GRS*EB | −0 (0) | GRS*EC | −0 (0) | GRS*BS | 0 (0) |
| | GRP*EB | 0.011 *** (0.003) | GRP*EC | 0.010 *** (0.003) | GRP*BS | 0.001 *** (0) |
| F-value | | 9.271 *** | | 4.433 *** | | 6.255 *** |
| R-squared | | 0.563 | | 0.381 | | 0.465 |

Notes: Dependent variable is RDA; the standard errors of the parameter assessments are included in brackets (); * $p < 0.1$, ** $p < 0.05$, *** $p < 0.01$.

RQ3 concerned the differential influences that corporate financial capability had on firms with different corporate governance structures. Multiple regression analyses were used to examine the research hypotheses and questions. The first column of Table 4 (model 4) tested whether the interaction between EB and corporate financial capabilities were significantly associated with firms' R&D intensity. The results indicated that the interaction explained 56.3% of the variance in firms' R&D intensity ($F$ (10, 72) = 9.271, $p < 0.01$). R&D intensity was positively related to the interaction variables between return on equity (ROE) and equity balance (EB), accounts receivable turnover (ART) and equity balance (EB), assets turnover (AT) and equity balance (EB), and profit growth rate (GRP) and equity balance (EB), but R&D intensity was negatively related to the interaction variables between cash flow rate (CFR) and equity balance (EB).

The second column of Table 4 (model 5) tested whether the interaction between EC and corporate financial capabilities was significantly related to firms' R&D intensity. The results indicated that the interaction explained 38.1% of the variance in firms' R&D intensity ($F$ (10, 72) = 4.433, $p < 0.01$). Additionally, R&D intensity was positively related to the interaction variables between return on equity (ROE) and equity concentration (EC), accounts receivable turnover (ART) and equity Concentration (EC), assets turnover (AT) and equity concentration (EC), and profit growth rate (GRP) and equity concentration (EC), but was negatively related to the interaction variables between return on assets (ROA) and board size (BS), leverage (LEV) and equity concentration (EC), and CFR and equity concentration (EC).

The third column of Table 4 shows the results of model 6 testing whether the interaction between BS and corporate financial capabilities was significantly related to firms' R&D intensity. The results indicated that the interaction explained 46.5% of the variance in firms' R&D intensity ($F$ (10, 72) = 6.255, $p < 0.01$). R&D intensity was positively related to the interaction variables between ROE and BS, ART and BS, AT and BS, and GRP and BS, but negatively related to the interaction variables between ROA and BS, LEV and BS, CFR and BS, and GPA and BS.

## 5. Discussion, Conclusions, and Suggestions

### 5.1. Discussion and Conclusions

This study empirically examined the effects of corporate governance structure and corporate financial capability on R&D intensity in the context of sports firms listed on The New Third Board in China. The results showed that corporate financial capability was a major determinant of R&D intensity, but that corporate governance structures had a very limited effect on R&D intensity.

To begin, it appeared apparent that profitability affected R&D intensity in Chinese sports firms. Particularly, return on equity significantly and positively affected R&D intensity, but there was no evidence to support the notion that return on sales and return on assets affected R&D intensity. This finding was different than that of Wang and You (2009) [83], who found that return on equity was not related to R&D investment. The average return on equity among Chinese sports firms was very small; so, when their return on equity would increase, shareholders' enthusiasm in R&D investment would likely increase as well. However, if their returns on sales and assets would increase, shareholders' enthusiasm in sales input and fixed asset investment would likely increase first.

Further, we found that solvency affected R&D intensity. Specifically, leverage negatively affected R&D intensity. This study had similar findings to those of Badia and Slootmaekers' (2009) [38] and Hillier et al.'s (2011) [34] studies, indicating that leverage affected the firms' capability to access liquidity. Although the mean of leverage was at a very appropriate range from 0.4 to 0.6, and many sports firms were able to raise funds with loans, leverage still negatively affected their R&D intensity. This showed that managers of Chinese sports firms had higher risk aversion and lower self-confidence with regard to R&D investment.

Additionally, this study found that operating capability affected R&D intensity. Asset turnover and accounts receivable turnover were positively related to R&D intensity, demonstrating that they were two important indicators of the financial security of the enterprise, the degree of capital preservation, and the profitability of the assets. However, R&D intensity was found to have a negative relationship with the cash flow ratio. These results contrasted with those derived by Himmelberg and Petersen (1994) [32], Hottenrott and Peters (2012) [84], and Sasidharan, Lukose, and Komera (2015) [85]. However, our results reflected those of Podolski (2016) [86] and Seifert and Gonenc (2012) [18], who found a similar negative relationship between cash flow and R&D investment. According to Dasgupta, Noe, and Wang (2011) [87], Driver and Guedes (2012) [47], and Huang and Wang (2015) [33], the negative relationship occurred because of the "pecking order" of cash flow use. That is, an immediate increase in cash flow is used to build cash stock for the short term and to reduce debt rather than to invest in projects like R&D [81,87]. Moreover, the higher adjustment costs of R&D projects may discourage managers from investing in them [32,88]. In order to fund R&D projects, it is more important to reduce financial debts of over both the short run and the long run than increase cash balances for firms [87,88].

Furthermore, growth positively affected R&D intensity; specifically, profit growth was positively related to R&D intensity, but sales growth and asset growth were not significantly related to R&D intensity. As an important aspect of corporate development, asset growth indicates the expansion speed of the asset management scale over a specific period. Sales growth is an important indicator for measuring the status of a firm's business and its ability to gain market share, and for predicting its development trends. High asset growth and high sales growth are attractive to external investors, but the external investment is curbed by low returns on assets and sales for Chinese sports firms. For

internal investors, high asset growth is difficult to use to further encourage R&D investment, given the low return on assets from R&D activities (i.e., long-term, high-risk investments with high uncertainty of return) [22,23]. Moreover, when firms' sales grow, their managers may invest in sales and production rather than R&D.

The effect of corporate governance structure on R&D intensity could not be neglected. Although its effect was not significant when only focusing on corporate governance structure but not on corporate financial capability (see model 2 in Table 3), its effect arose when corporate governance structure and corporate financial capability collectively took into effect. The evidence supported the conclusion of Vo and Le (2017) [17] that some indicators of financial capability drove the effects of corporate governance structure on R&D investment. Thus, corporate financial capability is an important mediator variable between corporate governance structure and R&D intensity, explaining an additional 3.1% of the variance in the relationship.

Continuing, equity balance was positively related to R&D intensity, but R&D intensity was not significantly related to equity concentration, the board size, or board independence. This differed with the results of Dong and Guo (2010) [9] and Driver and Guedus (2012) [47], as in this study, board independence was not found to directly influence R&D intensity in Chinese sports firms. We offered the following explanations: There were just so few sports firms (6.02%) with independent directors that the influence of independent directors on R&D investment did not materialize in our context. Consistent with the conclusions of Huang and Wang (2015) [33], the influence of board size on R&D intensity was not found. Similarly, most sports firms in China (72.3%) only had five shareholders and only met the minimum requirements of Corporate Law in China. Thus, the value of board size was unobvious for sports firms in China. Moreover, the average share rate held by the first major shareholder was more than 50% for Chinese sports firms, meaning that most firms were held by family owners. Higher shares held by families generally lead to decreasing levels of R&D investment, as family owners have higher risk aversion, and a large part of their wealth is invested in the firm [48]. Firms controlled by a dominant shareholder are more likely to pursue suboptimal, low-risk investments [33,89]. This explaining could be further confirmed by the Lazzarotti and Pelleprini (2015) [90] who thought that family firms by non-family managers were motivated by more aggressive and opener innovation strategy as compared to family firms by family managers. Thus, R&D intensity was positively related to equity balance.

Lastly, although R&D intensity was not significantly related to equity concentration or board size, the effect of corporate financial capability on R&D intensity was moderated by the influence of equity concentration and board size. By comparing model 1 with models 4, 5, and 6, we found that R&D intensity was negatively related to leverage in model 1; yet, R&D intensity was not significantly related to the interaction variable between leverage and equity balance in model 4. This showed that firms with a favorable equity balance more easily overcame financial constraints to increase R&D investment. As is consistent with the opinion of Stein (2003) [62], the influence of financial constraints would disappear in well-governed firms as investors would be more assured so the "lemons" problem, referring to issues that arise regarding that the value of an investment is underestimated or overestimated due to asymmetric information possessed by the buyer and the seller, is attenuated [62]. Moreover, R&D intensity was not significantly related to return on assets in model 1, but it was negatively related to the interaction variables between return on assets and equity concentration in model 5, and between return on assets and board size in model 6. These showed that for firms with excessive concentrations of equity, return on assets was more likely to affect their R&D investment. Dong and Guo (2010) [9] drew similar conclusions, arguing that the managerial discretion of CEOs was significantly and negatively related to R&D investment. It is necessary to note that R&D intensity was not found to be significantly related to asset growth in model 1; conversely, it was negatively related to the interaction variable between asset growth and board size. These suggested that firms with large board size, return on assets, and growth rate of assets were more likely to affect their R&D investments. The owners of firms

with lower returns on assets may have heightened fears and take fewer risks on R&D; whereas, board monitoring is less effective as the board gets larger [33].

In brief, these insights contribute to a more in-depth understanding of independent innovation in an emerging industry development context. Our findings could potentially provide practical guidance to both managers and government-industry policymakers in the sports industry.

*5.2. Suggestions*

Based on the observed effects of corporate financial capability and corporate governance structure on R&D intensity, Chinese sports firms should improve their governance structure and heighten their financial capability in the following ways as they strive to promote independent innovation. First, in addition to technological innovation, sports firms should attach increased importance to management innovation. Specifically, sports firms should remove constraints on R&D investment, heighten the efficiency of R&D activities, reduce the risk of R&D projects, and increase the self-confidence and enthusiasm of innovation. Meanwhile, sports firms could reduce their anxiety about R&D risk by actively buying technology insurance. Second, sports firms should improve their governance structure. Although the influences of equity concentration, the board size, and board independence on R&D intensity are not supported by our evidence, sports firms should moderately reduce the equity concentration. Specifically, they should break the family governance mechanism and build a corporate agency mechanism. Sports firms should further improve the equity balance mechanism by expanding the size of the board and attracting outside directors with backgrounds in technology. Thirdly, sports firms should improve their financial capabilities. Specifically, they should further increase the return on equity to heighten the investment enthusiasm and confidence of shareholders. They should also improve the contributions of R&D results to the returns on sales and assets by heightening the success rate of R&D projects and attaching importance to the R&D of high potential products. In addition, sports firms should increase the determination of R&D and lower the anxiety of leverage. Lastly, considering the interaction effect of financial capacity and governance structure, sports corporations with different financial capacity should adopt differential governance structures in an effort to enhance their innovative activities and market competitiveness to get ready for the ever-changing market environment.

In closing, the findings of this study demonstrated that corporate financial capability influenced R&D intensity in Chinese sports firms; yet, their effectiveness varied among sports firms with different governance structures. Corporate governance structure had a small but meaningful effect on R&D intensity, and its effect depended on corporate financial capability. In order to promote independent innovation, sports firms should improve their governance structure and financial health. The government should also provide guidance for the management innovation of sports firms through offering training, benchmarking, and evaluation programs. The findings of this study are deemed useful for sports business managers who wish to improve and expand upon their innovation-driven practices. These research findings were extensively discussed based on the related literature derived in a variety of business disciplines. To some extent, the research findings might have general applicability to the innovation management of new start-ups in addition to the Chinese sports industry.

This study paid attention to the issue of innovation development of firms from a new industry in China, a country with a growing economy and centralized administrative system, and was particularly focused on the effect of corporate financial capability and corporate governance structure on their R&D intensity. As the research sample came from the Chinese context, the research findings were somewhat delimited to this setting and generalizable to a similar market environment. Even so, the development of research hypotheses, selections of variables, and selection of the research protocol were heavily based on the research findings and researchers' indications that were derived from mainstream business studies and sports business studies conducted in many other countries located in both the West and East. It is reasonable to assume that the research findings could provide a meaningful reference for countries with a government that has a strong influence on the innovative development

of its businesses and, in particular, a strong desire to advance its sports industry through research development and technology innovations.

**Author Contributions:** Conceptualization, G.C.; Data Curation, G.C.; Formal analysis, G.C. and N.D.P.; Investigation, G.C.; Methodology, G.C.; Software, G.C.; Writing-Original Draft Preparation, G.C.; Wring-Review and Editing, G.C., J.J.Z., and N.D.P.; Project Administration, J.J.Z.; Supervision, J.J.Z.

**Funding:** This research was funded by the Chinese National Social Science Fund Project funding (No. 16BTY044) and Hubei Province Education Department Science Research Fund Project funding (No. B2016251).

**Conflicts of Interest:** The authors declare no conflict of interest.

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
