# Peer review of "Corporate Governance Structure, Financial Capability, and the R&D Intensity in Chinese Sports Sector: Evidence from Listed Sports Companies"

_sustainability, doi:10.3390/su11236810_

Round 1
Reviewer 1 Report
Review of “Corporate governance structure, financial capability, and the R&D intensity in Chinese sports sector: Evidence from listed sports companies”
The paper investigates the influence of corporate financial capability and corporate governance structure on firms’ R&D intensity through a series of multiple regression models. The research was carried out on Chinese sport firms listed in a Chinese stock exchange index (New Third Board).
The research treats an interesting thematic and discover some remarkable relationships. However, this review discovered some relevant weakness points in the paper that may be improved before the publication.
Hereinafter, the main weakness points of the paper are listed.
The first section is not well organized. The paper should starts with an introduction that provides the justification of the research. Currently, the section called research background explains something about the justification, but I think the justification should be better emphasize and isolated. Also the title of this section is quite misleading, because we expect the paper starts with an “introduction” to the work. It seems that the justification etc. are provided in the section 1.3 (after the hypotheses). I think you should move the concepts of section 1.3 to section 1.1 or, anyway, before the hypotheses. The section 1.2, where the hypotheses are proposed, may be moved and reallocated in a different section (a new “Section 2”). In this way, an entire section will be dedicated to the hypotheses. Given the affinity in the method used in the research (hypotheses + multiple regression model), I suggest to see the following reference for a possible paper structure: Antonacci, G., Fronzetti Colladon, A., Stefanini, A., & Gloor, P. (2017). It is rotating leaders who build the swarm: social network determinants of growth for healthcare virtual communities of practice. Journal of Knowledge Management, 21(5), 1218-1239. I suggest to include a figure (after the hypotheses) that summarize all the hypotheses/relationships analyzed by the paper. This can help the reader to quickly view/remember which are the hypotheses/relationships tested by the paper. I’m wondering if the section 2.2 should be posed before section 2.1 where the variables are putted in equations. Table 2 is superfluous/unnecessary. The main concept behind this table is already expressed in the text while it is unnecessary the details like turnovers etc. In the results the relationships with p<0.1 where reported and discussed, as variable EB – Hypothesis 7 and the variable ROA in model 5. A p-value bigger than 0.05 makes the test not reliable at all. A p-value between 0.05 and 0.01 is “valid” but not so strong. I think the discussion of these results (and the results themselves) should be readjusted coherently. Finally, I suggest to better generalize the research respect to the specific context chosen. Currently, the research appear to recall continuously the Chinese context. However, the research should generalize the results obtained in some way. Otherwise, the research limits its own validity. This point affect quite all the sections of the paper.
In addition, a short list of minor problems is provided:
“The New Third Board” is repeated too much time. It is interesting just when you describe the context, but I would avoid to repeat it so much time. The equations (2) and (4) are reported in a different graphical way respect to (1) and (3). “Document No. 46” is not well explained and probably it is not important for the paper.Author Response
Response to Reviewer 1 Comments
Point 1: The first section is not well organized. The paper should start with an introduction that provides the justification of the research. Currently, the section called research background explains something about the justification, but I think the justification should be better emphasize and isolated. Also the title of this section is quite misleading, because we expect the paper starts with an “introduction” to the work. It seems that the justification etc. are provided in the section 1.3 (after the hypotheses). I think you should move the concepts of section 1.3 to section 1.1 or, anyway, before the hypotheses. The section 1.2, where the hypotheses are proposed, may be moved and reallocated in a different section (a new “Section 2”). In this way, an entire section will be dedicated to the hypotheses. Given the affinity in the method used in the research (hypotheses + multiple regression model), I suggest to see the following reference for a possible paper structure: Antonacci, G., Fronzetti Colladon, A., Stefanini, A., & Gloor, P. (2017). It is rotating leaders who build the swarm: social network determinants of growth for healthcare virtual communities of practice. Journal of Knowledge Management, 21(5), 1218-1239. I suggest to include a figure (after the hypotheses) that summarize all the hypotheses/relationships analyzed by the paper. This can help the reader to quickly view/remember which are the hypotheses/relationships tested by the paper. I’m wondering if the section 2.2 should be posed before section 2.1 where the variables are putted in equations.
Response: According to your guidance, we have improved the structure of the paper. The introduction section has now been divided into two parts: (a) general introduction (i.e., rationale establishment for research background, research purpose, research thinking, and significance of study) and (b) theoretical framework and hypothesis development. We have added a figure to summarize all of the hypotheses. The orders of section 2.2 and section 2.1 have been switched.
Point 2: Table 2 is superfluous/unnecessary. The main concept behind this table is already expressed in the text while it is unnecessary the details like turnovers etc.
Response: According to your suggestion, we have deleted Table 2. Meanwhile, considering that the information in Table 2 can help readers understand that sport industry in China is considered a rising green industry, we have kept its content descriptions in the text. Table 3 in the previous version is a description of China's sport firms listed on the New Third Board, including information on turnovers, which helps lay a foundation for illustrating research findings and later, conducting discussions. Thus, we have kept Table 3 in the text.
Point 3: In the results the relationships with p<0.1 where reported and discussed, as variable EB – Hypothesis 7 and the variable ROA in model 5. A p-value bigger than 0.05 makes the test not reliable at all. A p-value between 0.05 and 0.01 is “valid” but not so strong. I think the discussion of these results (and the results themselves) should be readjusted coherently.
Response: For an applied study, we agree with you that an alpha level of 0.05 should be kept as the criterion for judging statistical significance. An alpha level of 0.01 is appropriate for a confirmatory study. In this study, we examined if corporate governance structure was significantly associated with firms’ R&D intensity as shown in Model 2 and Model 3 in Table 4. Model 2 explored the effect of corporate governance structure on R&D intensity, and its p-value is not only greater than 0.05, but also greater than 0.1 (i.e., p > 0.1). Thus, our conclusion is that corporate governance structure was not directly affecting R&D intensity, rejecting the related hypothesis.
To date, no previous study has been found that examined research and development issues in the Chinese sport industry; thus, to a great extent the current study was exploratory in nature. An alpha level of 0.10 may be adopted for an exploratory study as indicated by Alam, Uddin, and Yazdifar (2019) and Zhang and Guan (2018) who conducted studies on R&D in mainstream business settings. As shown in Model 3, EB has a weak, positive relationship with R&D intensity (p < 0.1). If not for adopting the standard of alpha = 0.10 for an exploratory study, this piece of the information would have gotten totally missed. Variance explained in Model 3 was more than that in Model 1, further suggesting that corporate governance structure had a small but meaningful impact on R&D intensity, which occurred in the presence of corporate financial capability.
Zhang, J. & Guan, J. (2018). The time-varying impacts of government incentives on innovation. Technological Forecasting & Social Change, 135, 132-144.
Alam, A., Uddin, M., Yazdifar, H.,(2019). Institutional determinants of R&D investment: Evidence from emerging markets. Technological Forecasting & Social Change, 138,34-44.
Point 4: Finally, I suggest to better generalize the research respect to the specific context chosen. Currently, the research appears to recall continuously the Chinese context. However, the research should generalize the results obtained in some way. Otherwise, the research limits its own validity. This point affect quite all the sections of the paper.
Response: This study has paid attention to the issue of innovation development of firms from a new industry in China, a country with a growing economy and centralized administrative system, and was particularly focused on the effect of corporate financial capability and corporate governance structure on their R&D intensity. As the research sample comes from the Chinese context, the research findings are somewhat delimited to this setting and generalizable to similar market environment. Even so, development of research hypotheses, selections of variables, and selection of the research protocol were heavily based on the research findings and researchers’ indications that were derived from mainstream business studies and sport business studies conducted in many other countries located in both the West and East. It is reasonable to assume that the research findings can provide a meaningful reference for countries with a government that has a strong influence on the innovative development of its businesses and in particular a strong desire to advance its sport industry through research development and technology innovations.
Point 5: In addition, a short list of minor problems is provided: “The New Third Board” is repeated too much time. It is interesting just when you describe the context, but I would avoid to repeat it so much time. The equations (2) and (4) are reported in a different graphical way respect to (1) and (3). “Document No. 46” is not well explained and probably it is not important for the paper.
Response: In this revision, we have substantially reduced the appearance frequency of “The New Third Board”. Thank you also for pointing out the typos in EQ (2) and EQ (4); accordingly, we fixed these.
“Document No.46” was a policy document released by the central government of China in 2014, aiming to rapidly develop the sport industry and promote sports consumption. It has been widely considered an impactful policy and initiative, in which innovation-driven measures are particularly highlighted. Since then, development of Chinese sport industry has substantially been escalated. Numerous new start-ups have emerged in the sport industry and increasingly more enterprises have incorporated technological innovations into their strategic planning and operational undertaking. Yet, smaller, newer, and more technology-intensive firms face financial and corporate governance constraints (Yung, 2016; Brown et al, 2012) despite recognizing the importance and relevance of research and development activities on the firm’s long-term well-being (Pham et al, 2018). Therefore, it is critical for us to discuss the effects of corporate finance capabilities and governance structure on R&D investment in the context of the Chinese sport industry.
Finally, we hope that we have adequately addressed your suggestions and concerns. Thank you for taking the time to review our manuscript. We truly believe that your constructive comments have helped improve the quality of our manuscript substantially.

Reviewer 2 Report
See report

Author Response
Response to Reviewer 2
Point 1: you need to justify why “China” why “sports industry”. Are they a unique experiment lab for answering the research questions?
Response: We have addressed your question from the following perspectives. First, the sports industry is a very important green industry for the sustainable development of mankind. Secondly, the innovation development of firms in a new industry is an important means to promote the rapid development of the industry, and it also is a very important academic issue. The sport industry in China precisely is a new rising industry. Based on the above two reasons, in this study we have investigated the “sport industry” and its innovation development issues.
An important reason why we paid attention to “China” is that China’s sport industry has unique characteristics. On one hand, there is strong realistic need for innovation development in the sport industry in China. Healthy growth of the sport industry is very important to the country’s economic and social development; yet, there is a large gap between the sport industry in China and that in some of the western countries. Innovation capability in sport firms is fundamental for jumpstarting the sport industry in China. On the other hand, there is strong external incentive to the innovation development of sport firms in China. The central government of China attaches importance to the development of sport industry and is currently implementing an important practice to support the innovation development of sport firms, which creates a unique research context to explore the enterprise level of innovation development for a new industry. The central government of China has introduced innovation-driven policies for the first time in Document No. 46. Partially attributing to this market environment, numerous new start-ups have emerged in the sport industry, and more and more of them have begun to put emphasis on nurturing the capacity of enterprise innovation. Although the Chinese government have promoted a creative business cultural and environment through the innovation-driven policies in an effort to advance the development of the sport industry in China, innovation-driven policy is merely an external incentive to affect R&D investment. In the meantime, sustainable development of a sport enterprise would greatly depend on many of its internal factors in finance and governance. Therefore, this paper selects “sport industry” and “China”, and explores the effect of financial capability and corporate governance structure on R&D intensity of sport firms.
Social, psychological, economical, and environmental impacts of organized sport activities and sport events are well-documented in previous studies (Savic et al., 2017). When planned and managed effectively, sports can be a green industry that positively promote public welfare of communities and citizens. Hence, the sport industry is a green industry that is very important for the sustainable development of mankind. However, there is still gap between the sport industry in China and that of some western countries. Due to growing interest in leisure activities, adoption of healthy lifestyles, augmentation of sport competitions and events, and evolution of sport media technology, the sport industry has experienced rapid growth and has become one of the largest industries in some western countries. For example, the estimated size of the sport business industry in North America has risen sharply to approximately $600 billion in 2018 (Plunkett Research, 2018). Sport leagues, teams, and events, the most popular leisure and entertainment options, have been integral to many communities’ cultural and economic foundations (Zhang, Cianfrone, & Sun, 2015). However, there lacks of such strong market demand and such competitive sport firms in China, and the total output of the sport industry in China was only 2.4 trillion RMB in 2018. The international nature of modern sport requires sport organizations to modify their management practices in order to remain effective and competitive in a border-transcended marketplace (Means & Nauright, 2007; Pfahl, 2011). Differences in such areas as culture, policy and regulation, language, and environment in global, national, regional, and local communities make this a challenging task. Thus, the rapid and sustainable development of Chinese sport industry need to be promoted by continual innovation.
In order to rapidly develop the sport industry, the central government of China released Document No. 46 titled “to rapidly develop sports industry and promote sports consumption” in 2014. Under its support, China’s sport sector has been growing at an annual rate of over 10% in recent years, with some sectors such as sport facility construction, sport management services, and fitness and leisure growing at a rate of over 50%. Obviously, the role of innovation of sport firms in the development in China’s sport industry cannot be overlooked. Especially, innovation-driven policies was also introduced for the first time in Document No. 46. Partially attributing to this market environment, numerous new start-ups have emerged in the sport industry, and more and more of them have begun to put emphasis on nurturing the capacity of enterprise innovation. A large amount of literature showed that many national or local governments routinely implement innovation-driven policies or other forms of stimulation to address hindrance associated with R&D underinvestment and stimulate creativity, advancement, and competitiveness, such as tax relief and deductions, R&D subsidies, and regulatory provisions (Brown, et al., 2017; Carboni, 2017).
Numerous researchers have indicated that research-based innovations are the foundation for a business corporation to survive, thrive, and sustain; sport businesses are no exception. Research & Development (R&D) has become a core business component of firms (Porter & Millar, 1985) as it is both a key ingredient in the introduction of new products and processes (Honore et al., 2015) and a key source of rapid growth, sustainable development, and competitive advantages (Dong & Gou, 2010). R&D investment plays a decisive role in initiating and sustaining innovation activities of technological enterprises (Lee, et al., 2018), not only improving short-term and long-term financial performance but also facilitating the acquisition of competitive advantages (Chintrakarn et al., 2016). Thus, it is very necessary to focus on the innovation development of sport firms, especially their R&D investment issue. Choosing the level of R&D as the criterion variable and identifying its predicting variables in this study would be ultimately for promoting sustainable development of sport enterprises in China.
Although the Chinese government have promoted a creative business cultural environment through the innovation-driven policies in an effort to advance the development of the sport industry in China, an innovation-driven policy is merely an external incentive to affect R&D investment. In the meantime, sustainable development of a sport enterprise would greatly depend on many of its internal factors in finance and governance. An enterprise’s financial capability has been recognized by previous researchers as the most pertinent determinant of its R&D investment, which can be facilitated or even deterred by the firm’s governance structure and its subsequent decision-making on R&D investment. As a rising industry, there are numerous new start-ups emerging in the sport industry. Yet, smaller, newer, and more technology-intensive firms face financial and corporate governance constraints (Brown et al, 2012; Yung, 2016) despite recognizing the importance and relevance of research and development activities for the firm’s long-term well-being (Pham et al, 2018). Today, many sport enterprises face financing and corporate governance constraints, which have hampered their market competitiveness, highlighting the significance of examining the influence of these concepts on research and development. Therefore, the purpose of this study was to examine the impact of corporate financial capacity and governance structure on the level of R&D investment of sport firms in the evolving and growing economic environment of China.
Point 2: you should clarify the contributions of the paper which are not elaborated well in the current paper. You can talk about the following contributions: What insights can you provide based on your finding? Do they push forward our understanding? What should we do with your research? Do you have any suggestions to improve the current regulation or practice? Adding the above discussion and extend your literature review may help you make more contributions and position your contributions better.
Response: We have made improvement according to your questions and guidance. This study was focused on the effects of corporate financial capability and governance structure on R&D investment in the context of the Chinese sport industry. Findings of this study potentially have the following implications pertaining to planning and executing R&D activities within sport companies: (a) identifying the importance and relevance of corporate financial capacity would help sport companies make appropriate decisions on investing into R&D activities and making choices among such options as self-innovation, purchasing intellectual property rights, and affiliating with larger corporations; (b) adopting a feasible governance structure would enhance the effectiveness of investing into and operating of R&D activities; (c) by exploring the interaction effect of financial capacity and governance structure on R&D intensity, suggestions can be made for corporations with different financial capacity to adopt differential governance structures in an effort to enhance their innovative activities and market competitiveness to get ready for the ever-changing market environment; and most importantly; and (d) this study paid a particular attention to the enterprise innovation in an new developing industry in a growing economy. Although the research sample comes from the Chinese sport industry, the research conclusion has some reference value for the innovation development of sport firms in other nations or geographical locations, even for the innovation development of new start-ups or new industry.
This study explored the effect of corporate financial capability and corporate governance structure on R&D intensity based on the evidence from Chinese sport firms listed on The New Third Board and found that corporate financial capability was an important determinate of R&D intensity. Corporate governance structure had a small but meaningful effect on R&D intensity and its effect depends on corporate financial capability. These research findings are extensively discussed based on the related literature derived in a variety of business disciplines. To some extent, the research findings may have general applicability to the innovation management of new start-ups in addition to the Chinese sport industry.
Point 3: To position your contributions better, you should link to more literature by discussing more relevant channels. You should consider, for example, market competition as a governance mechanism: Giroud, X., and H., Mueller, 2011, Corporate governance, product market competition, and equity prices. Journal of Finance 66, 563-600. The interactions between the executives, such as mutual monitoring among the executives: Li, Z.F., 2014, Mutual monitoring and corporate governance, Journal of Banking & Finance, 45, 255-269; Li, Z.F., 2018, Mutual monitoring and agency problem. https://www.researchgate.net/publication/272305464_Mutual_Monitoring_and_Agency_Problems; and external interactions between CEOs in the industry tournament: Coles et al. 2018, Industry Tournament Incentives, Review of Financial Studies, 31(4):1418-1459; On inside debt as governance: Li, F., Lin, S., Sun, S., Tucker, A. 2018. Risk-Adjusted Inside Debt. Global Finance Journal 35: 12-42. Or compensation incentives: Core, J. and Guay W., 1999, The use of equity grants to manage optimal equity incentive levels, Journal of Accounting and Economics 28, 151-184.
In addition, board diversity is also an important consideration. You need to discuss those aspects of possible channels to give readers a more comprehensive view and a richer story and/or point out future research direction from these perspectives.
Response: In this revision, we have made concerted efforts to address your comments and suggestions. In terms of the interactive effect of corporate governance structure and financial capability, we have discussed that corporate governance structure affects internal cash flow and external financing. Based on this premise, we have also made discussions on why and how corporate governance structure affect the performance of sport firms in section 2.3. According to previous research findings, mutual monitoring mechanism provides important checks and balances on CEO’s power and can mitigate the agency problem (Li, 2014), improve the value of firm, and reduce the CEO’s ability to pursue "quiet life” within a given context of corporate governance structure (Li, 2018). Moreover, higher fractional ownership, more optimal equity incentive levels, and more new equity incentives can lead to better firm performance (Morck, Shleifer, & Vishny,1988). Weak governance firms in noncompetitive industries have lower equity returns, worse operating performance, and lower firm value. Weak governance firms in noncompetitive situations have lower labor productivity and higher input costs than do good governance firms (Giroud & Mueller, 2011). To a varying degree, firm performance, market uncertainty, and investment risk of firms are associated with return-on-investment (Core & Guay, 1999); in the meantime, credit risk-adjusted inside the debt often motivates executives to be conservative (Jensen, & Meckling, 1976), which can be especially heightened when a credit crisis occurs in the marketplace (Li, Lin, Sun, & Tucker, 2018). The internal pay differential has a positive relationship with firm risk, R&D intensity, firm focus, and financial leverage (Kini & Williams, 2012). The industry pay gap affects firm performance, risk, investment policy, and financial policy, and the magnitude of its effects varies across industries and over time (Coles, Li, & Wang, 2018). These points are more clearly made in our revised manuscript.
Point 4: Minor Comments and Suggestions
There are many typos and grammatical mistakes throughout the paper, making it hard to read and understand. Try to avoid long sentences and vague words. Use short, precise, and concise sentences and be more straightforward. The last section should summarize all your findings, their implications to researchers and practitioners, future direction for research, limitation of the current study, etc. You need to seriously proofread the paper and extend and update your references.
Response: We are sorry about the typos and grammatical mistakes throughout the paper, and our revised manuscript have been checked by hiring a professional English editing service.
Accordingly, we have made major improvements in last section. The findings of this study demonstrate that corporate financial capability influences R&D intensity in Chinese sport firms; yet, their effectiveness varies among sport firms with different governance structures. Corporate governance structure had a small but meaningful effect on R&D intensity and its effect depends on corporate financial capability. In order to promote independent innovation, sport firms should improve their governance structure and financial health. Government should also provide guidance for management innovation of sports firms through offering training, benchmarking, and evaluation programs. The findings of this study are deemed useful for sport business managers who wish to improve and expand upon their innovation-driven practices. These research findings are extensively discussed based on the related literature derived in a variety of business disciplines. To some extent, the research findings may have general applicability to the innovation management of new start-ups in addition to the Chinese sport industry.
This paper paid attention to the issue of innovation development of firms in a new industry in China, a country with a growing economy and centralized administrative system, and was particularly focused on the effect of corporate financial capability and corporate governance structure on their R&D intensity. As the research sample comes from the Chinese context, the research findings are somewhat delimited to this setting and generalizable to similar market environment. Even so, development of research hypotheses, selections of variables, and selection of the research protocol were heavily based on the research findings and researchers’ indications that were derived from mainstream business studies and sport business studies conducted in many other countries located in both the West and East. It is reasonable to assume that the research findings can provide a meaningful reference for countries with a government that has a strong influence on the innovative development of its businesses and in particular a strong desire to advance its sport industry through research development and technology innovations.
Finally, we hope that we have adequately addressed your suggestions and concerns. Thank you for taking the time to review our manuscript. We truly believe that your constructive comments have helped improve the quality of our manuscript substantially.

Round 2
Reviewer 1 Report
Thank you for the kind answer.
The authors have improved the paper following the suggestions provided by the review. In my opinion, the paper can be now published.
Reviewer 2 Report
Well done. Congratulations!
This manuscript is a resubmission of an earlier submission. The following is a list of the peer review reports and author responses from that submission.
Round 1
Reviewer 1 Report
Referee Report
Main Comments and Suggestions
What specifically can we learn from 1. Chinese 2. Sports companies under the 3. regulation/incentive changes initiated by the government? You should clarify the contributions of the paper (in the first section “introduction” that you should separate from lit review) which are not elaborated well in the current paper. You can talk about the following contributions: What insights can you provide based on your finding? Do they push forward our understanding? What should we do with your research? Do you have any suggestions to improve the current regulation or practice? Adding the above discussion and extend your literature review may help you make more contributions and position your contributions better.
I find it interesting about the interactive effects of corporate governance. It is plausible that governance guarantees that good R&D projects are supported by and therefore are more responsive to financial capability. My main suggestion is that you should tell a richer story about corporate governance and link to more literature by discussing more relevant channels. You should consider, for example, market competition as a governance mechanism: Giroud, X., and H., Mueller, 2011, Corporate governance, product market competition, and equity prices. Journal of Finance 66, 563-600. The interactions between the executives, such as mutual monitoring among the executives: Li, Z.F., 2014, Mutual monitoring and corporate governance, Journal of Banking & Finance, 45, 255-269; Li, Z.F., 2018, Mutual monitoring and agency problem. https://www.researchgate.net/publication/272305464_Mutual_Monitoring_and_Agency_Problems; and external interactions between CEOs in the industry tournament: Coles et al. 2018, Industry Tournament Incentives, Review of Financial Studies, 31(4):1418-1459; On inside debt as governance: Li, F., Lin, S., Sun, S., Tucker, A. 2018. Risk-Adjusted Inside Debt. Global Finance Journal 35: 12-42. Or compensation incentives: Core, J. and Guay W., 1999, The use of equity grants to manage optimal equity incentive levels, Journal of Accounting and Economics 28, 151-184. You need to discuss those aspects of possible channels to give readers a more comprehensive view and a richer story and/or point out future research direction from these perspectives.
In conclusion, I would like to thank the authors for a very interesting, unique and potentially important paper. Hope these comments and suggestions can help further their study.
Author Response
GENERAL COMMENTS
What specifically can we learn from Chinese sports companies under the regulation/incentive changes initiated by the government? You should clarify the contributions of the paper (in the first section “introduction” that you should separate from lit review) which are not elaborated well in the current paper. You can talk about the following contributions: What insights can you provide based on your finding? Do they push forward our understanding? What should we do with your research? Do you have any suggestions to improve the current regulation or practice? Adding the above discussion and extend your literature review may help you make more contributions and position your contributions better.
Response: This study was focused on the effects of corporate financial capability and governance structure on R&D investment in the context of the Chinese sport industry. Findings of this study potentially have the following implications pertaining to the planning and executing R & D activities within sport companies: (a) identifying the importance and relevance of corporate financial capacity would help sport companies make appropriate decisions on investing into R&D activities and making choices among such options as self-innovation, purchasing intellectual property rights, and affiliating with larger corporations; (b) adopting a feasible governance structure would enhance the effectiveness of investing into and operating of R&D activities; (c) by recognizing the interaction effect of financial capacity and governance structure, suggestions can be made for corporations with different financial capacity to adopt differential governance structures in an effort to enhance their innovative activities and market competitiveness to get ready for the ever-changing market environment; and most importantly, (d) this study paid a particular attention to the enterprise innovation in an new developing industry in a growing economy. As the research samples come from the Chinese sport industry, the research conclusion has some reference value for the innovation development of sport firms that are still evolving and developing. In this revision, the above points have been added and made clear.
I find it interesting about the interactive effects of corporate governance. It is plausible that governance guarantees that good R&D projects are supported by and therefore are more responsive to financial capability. My main suggestion is that you should tell a richer story about corporate governance and link to more literature by discussing more relevant channels. You should consider, for example, market competition as a governance mechanism: Giroud, X., and H., Mueller, 2011, Corporate governance, product market competition, and equity prices. Journal of Finance 66, 563-600. The interactions between the executives, such as mutual monitoring among the executives: Li, Z.F., 2014, Mutual monitoring and corporate governance, Journal of Banking & Finance, 45, 255-269; Li, Z.F., 2018, Mutual monitoring and agency problem. https://www.researchgate.net/publication/272305464_Mutual_Monitoring_and_Agency_Problems; and external interactions between CEOs in the industry tournament: Coles et al. 2018, Industry Tournament Incentives, Review of Financial Studies, 31(4):1418-1459; On inside debt as governance: Li, F., Lin, S., Sun, S., Tucker, A. 2018. Risk-Adjusted Inside Debt. Global Finance Journal 35: 12-42. Or compensation incentives: Core, J. and Guay W., 1999, The use of equity grants to manage optimal equity incentive levels, Journal of Accounting and Economics 28, 151-184. You need to discuss those aspects of possible channels to give readers a more comprehensive view and a richer story and/or point out future research direction from these perspectives.
Response: In this revision, we have made concerted efforts to address your comments and suggestions. In terms of the interactive effect of corporate governance structure and financial capability, we have discussed that corporate governance structure affects internal cash flow and external financing. Based on this premise, we have also made discussions on why and how corporate governance structure affect the performance of sport firms. According to previous research findings, mutual monitoring mechanism improves the value of a firm and reduces the CEO’s ability to pursue “quiet life” (Li, 2014); higher fractional owner-ship, more optimal equity incentive levels, and more new equity incentives can lead to better firm performance (Giroud & Mueller, 2011; Morck et al., 1988). To a varying degree, firm performance, market uncertainty, and investment risk of firms are associated with return-on-investment (Coles, Li, Wang, 2018); in the meantime, credit risk-adjusted inside the debt often motivates executives to be conservative (Jensen & Meckling, 1976), which can be especially heightened when a credit crisis occurs in the marketplace (Li, Lin, Sun, & Tucker, 2018). These points are more clearly made in our revised manuscript.
Reviewer 2 Report
This paper examines the impact of corporate governance structure and financial capability on R&D intensity of sports firms in China. This paper has some drawbacks.
Motivation
1. This paper is poorly motivated. The link between motivation and research questions is very weak. The first sentence of abstract and introduction all described a series of innovation-driven policies in 2014 for the sports industry. It prompts me to think that this paper is positioned to evaluate the policy impact. However, other than the first sentences, there are no details on the series of policies. It is impossible to assess to what extent the policy is related to innovation, and what type of innovation the policy aims to achieve, and what form of support the government gives to the sport industry.
2. Without a good motivation, it is difficult to appreciate the importance of research questions. This is to say that when we take away the motivation and just to look at research questions, it is not clear why the authors want to examine the relationship of corporate governance to R&D in sports industry. Therefore, proper motivation is critical for us to understand the value of research questions and the contribution to the literature.
Methodology, Data and Sample
3. If the authors want to examine the policy impact. Then the appropriate method should be a difference-in-difference model.
4. The source of the data is not disclosed. The number of observations and sampling period are not mentioned. The sample firms include firms in fitness training, sport media, etc. It seems that those type of companies does not need innovation policies to grow. A more detailed statistics are required in order to understand the nature of R&D activities for the firms in sport industry.
5. The context of this paper seems unrelated to the sustainability issue. It poorly fits into the scope of Sustainability.
Author Response
GENERAL COMMENTS
This paper is poorly motivated. The link between motivation and research questions is very weak. The first sentence of abstract and introduction all described a series of innovation-driven policies in 2014 for the sports industry. It prompts me to think that this paper is positioned to evaluate the policy impact. However, other than the first sentences, there are no details on the series of policies. It is impossible to assess to what extent the policy is related to innovation, and what type of innovation the policy aims to achieve, and what form of support the government gives to the sport industry.
Response: We are sorry about the confusion in our earlier writing, which apparently led to your misunderstanding. The primary purpose of our study was not about assessing the effectiveness of a policy; rather, we set out to examine the impact of corporate financial capacity and governance structure on the level of R&D investment of sport firms in the evolving and growing economic environment of China. In this revision, we have made strong efforts to alleviate this confusion.
It is our intention to describe the important research context of Chinese government having implemented an innovation-driven policy to support the development of the sport industry. Partially attributing to this market environment, enterprises have begun to pay attention to their capacity in technology innovation. Based on the literature that we have reviewed, many national or local governments routinely implement innovation-driven policies or other forms of stimulations to address R&D underinvestment, such as tax relief and deductions, R&D subsidies, and regulatory provisions (Brown et al., 2017; Carboni, 2017). The fundamental purpose of implementing innovation-driven policies in various countries is to solve issues resulted from R&D underinvestment by corporations and stimulate creativity, advancement, and competitiveness. Even so, an innovation-driven policy is only an external incentive to affect R&D investment.
Whereas, an enterprise’s financial capability has been recognized by previous researchers as the most pertinent determinant of its R&D investment, which can be facilitated or even deterred by the firm’s governance structure and its subsequent decision-making on R&D investment. With a supportive governmental policy for entrepreneurship in China, there have been increasingly more enterprises appearing in the sport industry; yet, their financial capabilities are overall weak and their governance structures appear less perfect, which have hampered their market competitiveness, highlighting the significance of investing into technology and product innovations.
In this revision, we have made clear that the new policy is merely a contextual information, not a focus of our investigation. Nonetheless, we have made suggestions for future investigations into the overall effect of the new policy in the discussion section of the manuscript.
Without a good motivation, it is difficult to appreciate the importance of research questions. This is to say that when we take away the motivation and just to look at research questions, it is not clear why the authors want to examine the relationship of corporate governance to R&D in sports industry. Therefore, proper motivation is critical for us to understand the value of research questions and the contribution to the literature.
Response: Please see our response above. This study was focused on examining the effects of corporate financial capability and governance structure on R&D investment in the context of the Chinese sport industry. Findings of this study potentially have the following implications pertaining to the planning and executing of R&D activities within sport companies: (a) identifying the importance and relevance of corporate financial capacity would help sport companies make appropriate decisions on investing into R&D activities and making choices among such options as self-innovation, purchasing intellectual property rights, and affiliating with larger corporations; (b) adopting a feasible governance structure would enhance the effectiveness of investing into and operating of R&D activities; (c) by recognizing the interaction effect between financial capacity and governance structure, suggestions can be made for corporations with different financial capacity to adopt differential governance structures in an effort to enhance their innovative activities and market competitiveness to get ready for the ever-changing market environment; and most importantly, (d) this study paid a particular attention to the enterprise innovation in an new developing industry and in a growing economy. As the research samples come from the Chinese sport industry, the research conclusion has some reference value for the innovation development of sport firms that are still evolving and developing. In this revision, the above points have been added and made clear.
Methodology, Data and Sample
If the authors want to examine the policy impact. Then the appropriate method should be a difference-in-difference model.
Response: Please see our Reponses above.
The source of the data is not disclosed. The number of observations and sampling period are not mentioned. The sample firms include firms in fitness training, sport media, etc. It seems that those type of companies does not need innovation policies to grow. A more detailed statistics are required in order to understand the nature of R&D activities for the firms in sport industry.
Response: The sample data come from the 2017 annual report of sport firms listed on the new enterprise board in China, which was actually publicized in 2018. In this report, there were a total of 83 sport firms, which became our research sample. Among these firms were mainly sporting goods manufactures, sport field construction companies, e-sports companies, sport media, and software development companies that heavily relied on entrepreneurship, technology, and innovation for development and sustainability. In addition to these firms, there were three sport clubs, seven health-fitness center, and five sport equipment sales firms, which carried different practices of innovational activities. For example, one health-fitness center provided sport medicine service and the sport equipment sales firms offered technical support to customers.
The context of this paper seems unrelated to the sustainability issue. It poorly fits into the scope of Sustainability.
Response: Numerous researchers have indicated that research-based innovations are the foundation for a business corporation to survive, thrive, and sustain; sport businesses are no exception. Sustainable development for a firm in the evolving Chinese sport industry is a particularly relevant and important issue to study. Social, psychological, economical, and environmental impacts of organized sport activities and sport events are well-documented in previous studies. When planned and managed effectively, sports can be a green industry that positively promote public welfare of communities and citizens. Choosing the level of R&D as the criterion variable and identifying its predicting variables in this study were ultimately for promoting sustainable development of sport enterprises in China, which would be consistent with the aims of the Sustainability journal. In this revision, we have made these points clear.
Thank you for your time and efforts to review our manuscript. We appreciate your constructive suggestions and criticism. We hope that we have adequately addressed your concerns.
Round 2
Reviewer 1 Report
well done
Reviewer 2 Report
I have read the “response to the reviewer’s comment” and revised form of the manuscript.
The revised manuscript does not significantly improve this paper. In addition, I find a new problem. According to the authors, the sample data is from 2017’s annual report, and there were a total of 83 firms used in this study.
There are three issues related to this sample. First, the number of observations. With only one year of data for 83 observations in a regression analysis, the statistic result is not robust and reliable. In corporate finance, it is common to use panel dataset to examine issues. I have concerns about results generated from such a small dataset. Furthermore, not every company will disclose the R&D expense, so the actual observations used in regression may be even lower than 83.
Second, what are those 83 firms in Third Board in China classified as sports firms? There are three industry classifications used in China.
CSRC industry classification 2001 edition CSRC industry classification 2012 edition Old industry classification (which is very broad with only six industry categories)I only find that there is “sport” mentioned in CSRC classification (2012) under “Culture, Sport & Entertainment” industry. There is only one company that is classified as “Sport” in this category and the company stock code is 002858. But this is not listed under Third Board. This information is sourced from CSMAR, a leading database provider for Chinese listed firms.
Third, the author claimed that sample firms are mainly sporting goods manufactures, sports field construction companies, e-sports companies, sports media, and software development companies. It is not clear whether sporting goods manufacturers also make other industrialized goods. If so, then the R&D expenditure they spend may not for “sporting” purpose, but for other purposes. E-sports companies are specialized in selling sports goods. The likelihood to have R&D expenditure is also very low.
In summary, I find that the empirical part of the paper is a very week point after the authors disclose more information about the data and sample.